# Scutellarein Inhibits LPS-Induced Inflammation through NF-κB/MAPKs Signaling Pathway in RAW264.7 Cells

**DOI:** 10.3390/molecules27123782

**Published:** 2022-06-12

**Authors:** Min Yeong Park, Sang Eun Ha, Hun Hwan Kim, Pritam Bhagwan Bhosale, Abuyaseer Abusaliya, Se Hyo Jeong, Joon-Suk Park, Jeong Doo Heo, Gon Sup Kim

**Affiliations:** 1Research Institute of Life Science, College of Veterinary Medicine, Gyeongsang National University, Jinju 52828, Korea; lilie17@daum.net (M.Y.P.); sangdis2@naver.com (S.E.H.); shark159753@naver.com (H.H.K.); shelake.pritam@gmail.com (P.B.B.); yaseerbiotech21@gmail.com (A.A.); tpgy123@gmail.com (S.H.J.); 2Biological Resources Research Group, Gyeongnam Department of Environment Toxicology, Chemistry, Korea Institute of Toxicology, 17 Jegok-gil, Jinju 52834, Korea; jdher@kitox.re.kr; 3Preclinical Research Center, Daegu-Gyeonbuk Medical Innovation Foundation (DGMIF), 80 Cheombok-ro, Dong-gu, Daegu 41061, Korea; jsp@kmedihub.re.kr

**Keywords:** scutellarein (SCU), LPS-induced inflammation, NF-кB, MAPK

## Abstract

Inflammation is a severe topic in the immune system and play a role as pro-inflammatory mediators. In response to such inflammatory substances, immune cells release cytokines such as tumor necrosis factor-α (TNF-α) and interleukin-1β (IL-1β). Lipopolysaccharide (LPS) is known as an endotoxin in the outer membrane of Gram-negative bacteria, and it catalyzes inflammation by stimulating the secretion of inflammatory-mediated cytokines such as cyclooxygenase-2 (COX-2) and inducible nitric oxide synthase (iNOS) by stimulated immune cells. Among the pathways involved in inflammation, nuclear factor kappa (NF-кB) and mitogen-activated protein kinases (MAPKs) are important. NF-kB is a diploid composed of p65 and IkBα and stimulates the pro- gene. MAPKs is a family consisting of the extracellular signal-regulated kinase (ERK), c-Jun NH2-terminal kinase (JNK), and p38, JNK and p38 play a role as proinflammatory mediators. Thus, we aim to determine the scutellarein (SCU) effect on LPS stimulated RAW264.7 cells. Furthermore, since scutellarein has been shown to inhibit the SARS coronavirus helicase and has been used in Chinese medicine to treat inflammatory disorders like COVID-19, it would be required to examine scutellarein’s anti-inflammatory mechanism. We identified inflammation-inducing substances using western blot with RAW264.7 cells and SCU. And we discovered that was reduced by treatment with SCU in p-p65 and p-IκBα. Also, we found that p-JNK and p-ERK were also decreased but there was no effect in p-p38. In addition, we have confirmed that the iNOS was also decreased after treatment but there is no change in the expression of COX-2. Therefore, this study shows that SCU can be used as a compound to treat inflammation.

## 1. Introduction

Inflammation is a complex response in living tissue that protects our bodies by neutralizing dangerous germs and preventing infection and wound healing [1]. In response to infection, innate immune cells such as fibroblasts, macrophages, mast cells, and neutrophils are activated. Pro-inflammatory cytokines by activated macrophages complicate many inflammatory diseases, reactive oxygen species (ROS), IL-1β, TNF-α, and IL-6 are important mediators [2]. Lowering these cytokines and pro-inflammatory agents may reduce inflammation.

LPS is one of the most effective macrophage activators [3]. LPS activates inflammatory mediators such as nitric oxide (NO) and prostaglandin E_2_ (PGE_2_) and induces overproduction. LPS stimulates iNOS and COX-2 synthesis by activating transcription factor NF-кB and activator protein (AP)-1. When transcription factors are activated and translocated to the nucleus in response to inflammatory stimuli, genes like iNOS and COX-2 are upregulated [4].

LPS-activated macrophages generate various inflammatory mediators, including NO, IL-6, IL-1, TNF-α, and PGE_2_, all contribute to host survival after infection and are essential for animals’ innate immune responses. COX-2 is assumed to be responsible for the synthesis of PGE_2_ in a range of inflammation models, whereas iNOS can boost NO production during inflammation [5]. Atherosclerosis cerebral malaria, rheumatoid arthritis, Parkinson’s disease, diabetes, and Alzheimer’s disease have all been linked to inflammatory conditions. As a result, in the treating of a range of inflammatory disorders, regulating macrophage-mediated inflammatory responses will be important [6].

MAPKs and NF-кB signaling pathways mediate inflammation. The main components of the NF-кB heterodimer are p65 and p50 [7]. After being liberated from IкB-α, they translocate from the cytoplasm to the nucleus. Both phosphorylation and degradation of IкB-kinase (IKK) play a role in NF-кB activation [7,8]. NF-кB translocates into nuclei after activation to stimulate the production of pro-inflammatory genes such as IL-1, IL-6, TNF-α, and iNOS [9]. ERK 1/2, p38, and JNK are members of the MAPKs family, which control the transducer to transmit environmental stimuli to the nucleus. The most closely associated with the pro-inflammatory mediators are JNK and p38 [10]. As a result, anti-inflammatory therapeutic strategies that target the NF-кB and MAPKs pathways may be intriguing.

Flavonoids are found in plants however glycosides are frequent [4]. Flavonoids, anthocyanins, tannins, catechins, isoflavones, lignans, and other polyphenol compounds can be found in large amounts in fruits and green leafy vegetables [4]. Many of the hydroxyl groups (–OH) found in polyphenols have good anti-inflammatory properties due to their ability to combine with other chemicals quickly [11,12,13].

Scutellarein (SCU) is a flavonoid monomer compound isolated from the medicinal herbs *Scutellaria barbata D. Don* and Erigeron breviscapus (vant) Hand Mass [14]. *Scutellaria baicalensis* is one of the most important herbs in traditional Chinese medicine, with a wide range of biological actions including anti-inflammation and anti-diarrheal properties [15,16]. Furthermore, anti-migratory, anti-proliferative, and apoptotic effects of SCU have been demonstrated in several human malignancies [14,17,18].

This study aims to investigate the effects of SCU in NF-кB and MAPKs pathways in LPS-activated RAW264.7 macrophage cells.

## 2. Results

### 2.1. The Effect of SCU on the Cell Viability of RAW264.7 Cells

Cell viability assay was performed on RAW264.7 cell with SCU. The cell toxicity was using the 3-(3,4-dimethyl-thiazolyl-2)-2,5-diphenyl tetrazolium bromide (MTT) assay. In this Figure 1B,C, the survival rate was 80% up to 75 μM/mL at both time and found no toxicity.

In the following tests, 25, 50, and 75 μM doses of SCU were used and 1 μg/mL concentration of LPS was used to stimulate. In addition, we fixed the 48 h treatment in further experiments.

### 2.2. Effects of SCU on iNOS and COX-2 Expression of LPS-Induced RAW264.7 Cells

The iNOS and COX-2 are involved in synthesizing NO, an inflammatory mediator [19]. The anti-inflammatory impact was investigated using western blot to determine the iNOS and COX-2 protein expression and we found that SCU suppresses iNOS expression in a dose-depencent manner whereas, it does not suppress the expression of COX-2, in RAW264.7 cells (Figure 2).

### 2.3. Inhibition of LPS-Induced NF-кB Pathways Activation by SCU

As a result, a western blot was used to investigate the SCU effect on NF-кB expression. We confirmed that p-p65 and p-IкBα were decreased when SCU was treated with LPS-induced RAW264.7 cells. (Figure 3). Treatment with SCU at concentrations at 25, 50, and 75 μM significantly reduced expression at p-p65 and p-IкBα. We also determined molecular docking between NF-κB and SCU. When molecular docking was performed, it was discovered that the SCU binds to 19 different sections (JNK 1, ARG 32, GLU 115, LEU 116, ASN 117, GLU 118, PHE 146, PRO 147, CYS 149, PHE 163, SER 164, HIS 183, GLU 184, TYR 227, GLU 233, ALA 234, LEU 236, ARG 237, ARG 239, ASN 240 and ZN 401).

### 2.4. Inhibition of LPS-Induced MAPKs Pathways Activation by SCU

MAPKs (JNK, ERK, and p38) are present in the cytoplasm without phosphorylation, but they become phosphorylated and translocated into the nucleus when stimulated by LPS (Figure 4). p38 appeared to have no effect SCU (not shown). Therefore, the phosphorylation alteration of MAPKs increased by LPS was validated by western blot to explore the influence of SCU on the activation of phosphorylated MAPKs.

## 3. Discussion

Inflammation is a complicated set of processes that allow tissues to respond to injury, and it necessitates the participation of different cell types that express and sequentially react to various mediators [20]. On LPS-stimulated RAW264.7 macrophages, we demonstrated the anti-inflammatory properties of scutellarein (SCU) in this current study.

iNOS is involved in NO generation, and iNOS suppression has been proposed as a potential anti-inflammatory target. In addition to NO, COX-2 produces PGE_2_, which is linked to the advancement of chronic inflammatory disorders [2]. As a result, decreasing NO and PGE_2_ production by inhibiting iNOS and COX-2 expression, respectively, has been suggested as a critical target for the treating inflammatory diseases [21]. So, in our study, COX-2 remains constant however iNOS tended to decrease with the concentration of drug SCU. Thus, the expression level of COX-2 remains the same, but the SCU reduces iNOS, suggesting that SCU may contribute to alleviating inflammation.

The protein complex NF-кB, which regulates DNA transcription, might bind to the NF-кB target site and increase transcription [22]. In reaction to inflammation, the p-IкBα protein activates NF-кB, which is ultimately destroyed by the proteasome. NF-кB is then phosphorylated and translocated into the nucleus, where it controls proinflammatory responses [23]. Reduced inflammation is linked to the reduction of NF-кB activation. Our findings prove that SCU rehabilitates RAW264.7 cells against inflammation by decreasing NF-кB activity. We also discovered that NF-кB and SCU were bound via molecular docking.

MAPKs proteins, which in macrophages include the ERKs and two stress-activated protein kinase (SAPK) families, JNK is an important upstream component of NF-кB [24]. MAPKs inhibitors have attracted a lot of attention because they’ve been related to important regulators of pro-inflammatory cytokine production. Therefore, in our study, we can check whether MAPKs components decreased in a dose-dependent manner. As a result, anti-inflammation is also conceivable due to the ability to block these MAPKs. Although p38 did not show any significant effect.

SCU demonstrated anti-inflammatory potential in LPS-induced RAW264.7 cells by regulating COX-2, iNOS, and cytokines, downregulating MAPKs, and blocking the NF-кB pathway. Furthermore, SCU can be reported to be efficacious in vivo [25]. Therefore, we suggest that SCU has the potential to be a therapeutic agent to relieve inflammation (Figure 5).

## 4. Materials and Methods

### 4.1. Chemicals and Reagents

Scutellarein (SCU) was purchased from Chengdu Biopurify Phytochemicals Ltd. (purity: >98%, Chengdu, Sichuan, China). 3-(4,5-Dimethylthiazol-2-yl)-2,5-diphenyltetrazolium bromide (MTT) was purchased from Duchefa Biochemie (Haarlem, The Netherlands). Antibodies to COX-2 (cat. no. 12282S), iNOS (cat. no. 13120S), p65 (cat. no. 8242S), phosphorylated p65 (p-p65) (cat. no. 3033S), IкBα (cat. no. 4812S), phosphorylated IкBα (p-IкBα) (cat. no. 2859S), JNK (Jun N-terminal kinase) (cat. no. 9258S), phosphorylated JNK (p-JNK) (cat. no. 4671S), ERK (Extracellular-signal-regulated kinase) (cat. no. 4695S), phosphorylated ERK (p-ERK) (cat. no. 4370S), p38 (cat. no. 8690S), phosphorylated p38 (p-p38) (cat. no. 4511S), and β-actin (cat. no. 4970S) were purchased from Cell Signaling Technology (Danvers, MA, USA). Horseradish peroxidase (HRP)-conjugated secondary antibodies to antirabbit (cat. no. A120-101P) and antimouse (cat. no. A90-116P) were obtained from Bethyl Laboratories, Inc. (Montgomery, AL, USA).

### 4.2. Cell Culture and Scutellarein (SCU) Treatment

The RAW264.7, mouse macrophage cells, were grown in full DMEM with 10% heat-inactivated fetal bovine serum (FBS) with 100 U/mL penicillin and 100 g/mL streptomycin. The cells were incubated at 37 °C in a humidified environment with 5% CO_2_. After seeding the cells, scutellarein (SCU) was added at a concentration of 25, 50, and 75 μM with LPS the next day. And incubate in an incubator for 48 h.

### 4.3. Cell Viability Assay

RAW264.7 cells were seeded at a density of 1.5 × 10^4^ cells per well in 96 well plates and then cultured, with LPS (1 µg/mL) and co-treatment with various concentrations of SCU (0, 25, 50, 75, 100, and 125 µM) at 37 °C for 48 h. After incubation, MTT solution (10 µL; 5 mg/mL) was added to the plate and incubated at 37 °C for ~2 h. The insoluble formazan crystals were then dissolved in DMSO after the growth media was entirely washed away. And the absorbance of the converted dye was measured at a wavelength of 560 nm by microplate reader Multiskan FC (Thermo Scientific, Rockford, IL, USA).

### 4.4. Western Blot

For western blot analysis, RAW264.7 cells were seeded into 60 mm plates at a density of 1 × 10^6^ cells/well and treated with 25, 50, and 75 µM SCU for 48 h at 37 °C. Then the cells were lysed in ice-cold RIPA buffer (50 mM Tris-HCL (pH 8.0), 0.5% sodium deoxycholate, 1 mM EDTA, 150 mM NaCl, 0.1 SDS and 1% NP-40). Protein concentrations were determined using the Pierce^TM^ BCA Protein Assay (Thermo Scientific, Rockford, IL, USA) according to the manufacturer’s instructions. Equal amounts of protein (10 μg) were separated via SDS-PAGE on 10% gels and transferred onto PVDF membranes using the JP/WSE-4040 HorizeBLOT 4M-R WSE-4045 (ATTO Blotting System, Tokyo, Japan). The blots were then blocked with EzBlockChemi (ATTO Blotting System, Tokyo, Japan) for 1 h at room temperature. Membranes were further incubated with 1:1000 dilutions of primary antibodies overnight at 4 °C. The membranes were washed three times for 10 min with TBS-T and probed with a second antibody until 2 h at room temperature. The second antibody was diluted at 1:5000. The blots were visualized using Clarity™ ECL substrate reagent (Bio Rad Laboratories, Inc., Hercules, CA, USA) and quantified by densitometry using ImageJ software (U.S. National Institutes of Health, Bethesda, MD, USA) with β-actin as the loading control. The experiment was performed in triplicate.

### 4.5. Molecular Docking

The structure of NF-κB was obtained at high resolution from a protein data bank (PDB) (https://www.rcsb.org/, accessed on 28 April 2022) with PDB ID 4Q3J, and the three-dimensional structures of the compound SCU were obtained from PubChem (https://pubchem.ncbi.nlm.nih.gov/, accessed on 28 April 2022) with PDB CID 5281697. The protein and ligand were docked using the USCF Chimera tool, and all possible conformations were returned using default parameters. The estimated free energy of binding and total intermolecular energy was used to evaluate the results.

### 4.6. Statistical Analysis

The data are expressed as the mean ± SEM. Data were analyzed using GraphPad Prism software (version 9.3.1; GraphPad Software, Inc.). Significant differences between groups were calculated by one-way factorial analysis of variance (ANOVA), followed by a Dunnett’s multiple comparisons test, and *p* < 0.05 was considered statistically significant.

## 5. Conclusions

We confirmed that SCU inhibited involved inflammation and blocks NF-κB and MAPK, which are pathways in which inflammation. SCU is also effective in down regulation of iNOS, suggesting that SCU is effective in inflammation. Consequently, SCU acts as an anti-inflammation agent by blocking the NF-κB and MAPKs pathways.

## Figures and Tables

**Figure 1 molecules-27-03782-f001:**
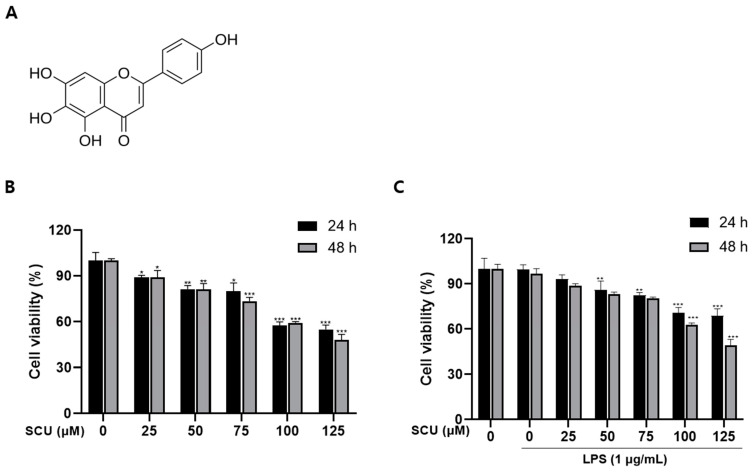
Cytotoxic effect of SCU on RAW264.7 cells treated with or without LPS (1 μg/mL) and treated with SCU(0, 25, 50, 75, 100, and 125 μM) at 37 °C for 24 h and 48 h. (**A**) Chemical structure of SCU. (**B**) SCU (0, 25, 50, 75, 100, and 125 μM) was treated by concentration, and then the toxicity of SCU to cells for 24 h and 48 h were measured. (**C**) Cell viability when LPS and SCU were treated together for 24 h and 48 h. Comparison with SCU and LPS treated group * *p* < 0.05, ** *p* < 0.01, *** *p* < 0.001.

**Figure 2 molecules-27-03782-f002:**
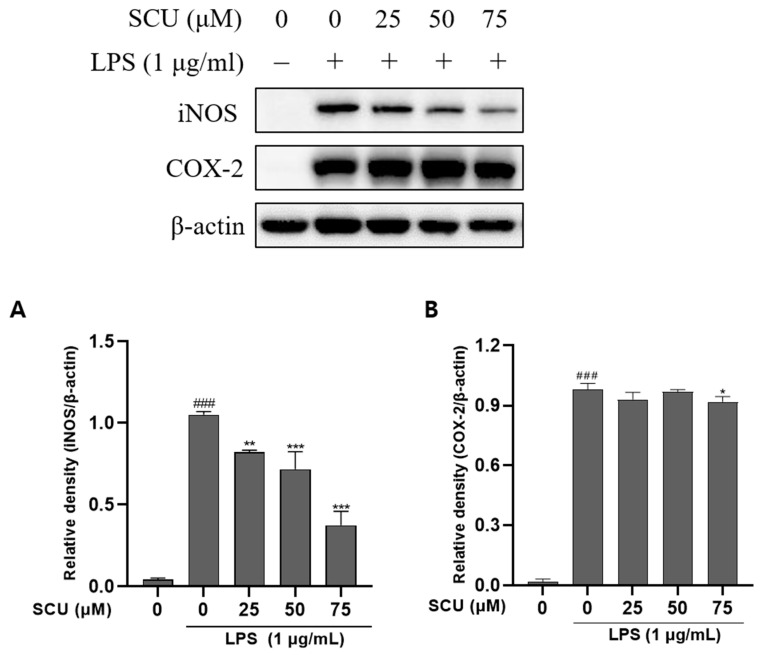
Effect of SCU on LPS-induced protein expression of iNOS and COX-2. (**A**) Relatibve density of iNOS expression, despite (**B**) Relative density of COX-2 expression. Comparison with only LPS treated ### *p* < 0.001. Comparison with SCU and LPS treated group * *p* < 0.05, ** *p* < 0.01, *** *p* < 0.001.

**Figure 3 molecules-27-03782-f003:**
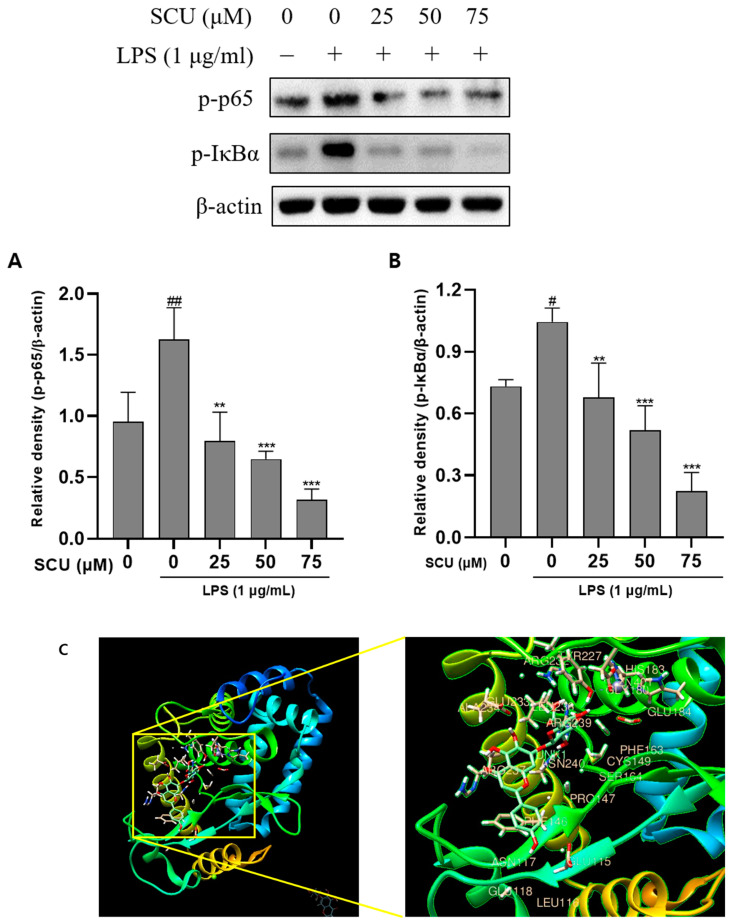
Effect of SCU on LPS-induced protein expression of p-p65 and p-IкBα. RAW264.7 cells were treated with LPS (1 μg/mL) and treated with SCU (0, 25, 50, and 75 μM) for at 37 °C 48 h. (**A**) Relative density of p-p65 (**B**) Relative density of p-IkBα. Comparison with only LPS treated group # *p* < 0.05, ## *p* < 0.01.** *p* < 0.01, *** *p* < 0.001. Comparison with SCU and LPS treated group (**C**) Molecular docking with NF-κB and SCU.

**Figure 4 molecules-27-03782-f004:**
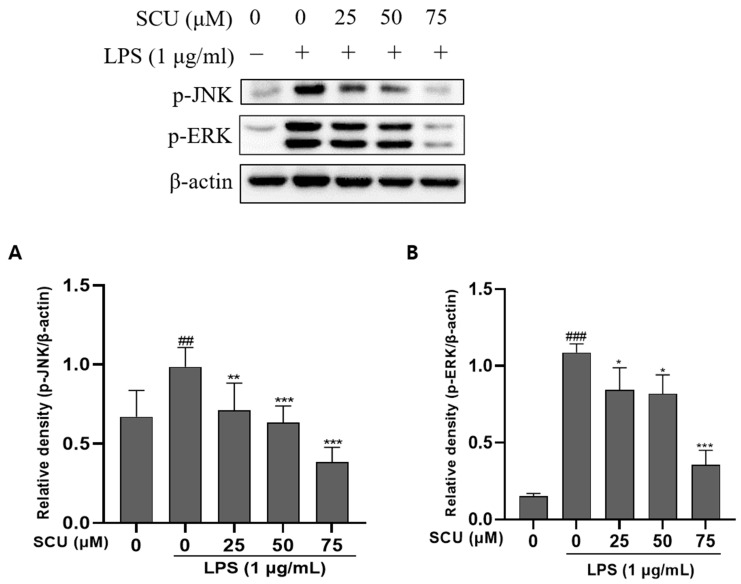
Effect of SCU on LPS-induced MAPKs protein expression. (**A**) Expression of p-JNK affected by SCU. (**B**) Expression of p-ERK affected by SCU. Comparison with only LPS treated group ## *p* < 0.01, ### *p* < 0.001. * *p* < 0.05, ** *p* < 0.01, *** *p* < 0.001.

**Figure 5 molecules-27-03782-f005:**
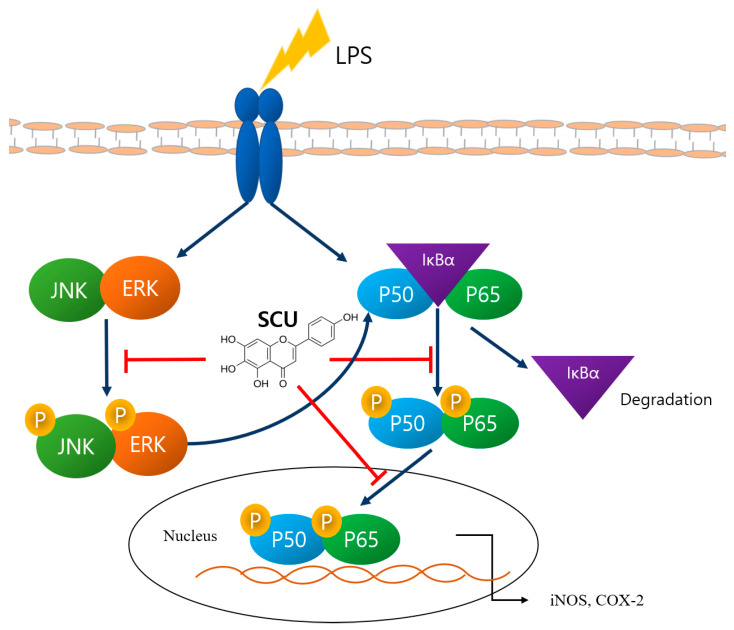
Summary of pathways in which the SCU effect.

## Data Availability

The data presented in this study are available on request from the corresponding author.

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
