# Peer review of "Scutellarein Inhibits LPS-Induced Inflammation through NF-κB/MAPKs Signaling Pathway in RAW264.7 Cells"

_molecules, 2022, doi:10.3390/molecules27123782_

Round 1
Reviewer 1 Report
The present article declared the scutellarein (SCU) effect on LPS stimulated RAW264.7 cells. They found that in p-p65, p-IκBα , p-JNK and p-ERK were down-regulated, but there was no effect in p-p38. The iNOS was decreased after treatment but there is no change in the expression of COX-2. Except some writing organization in the abstract, the paper is well organized and presented. Suggestions are listed as follows:
In the abstract, the background part could be shorter, like the section of “Inflammation is a severe topic ……………….play a role as pro-inflammatory mediators.” Instead, since scutellarein was found to be active on inhibiting SARS coronavirus helicase, and existed in some Chinese medicine that could be used to against other inflammatory relevant disease as Covid-19, so it would be nesseray to verify the pathway of anti-inflammatory of scutellarein. Therefore, a sentence for the hypothesis or aim could be added in the abstract, such as: To elucidate the if SCU can be used as an inflammation inhibitor and the possible inhibition pathway, the scutellarein (SCU) effect on LPS stimulated RAW264.7 cells was investigated.
And then, the last sentence of “Therefore, this study shows that SCU can be used as a compound to treat inflammation.”, could be deleted.
Author Response
The present article declared the scutellarein (SCU) effect on LPS stimulated RAW264.7 cells. They found that in p-p65, p-IκBα , p-JNK and p-ERK were down-regulated, but there was no effect in p-p38. The iNOS was decreased after treatment but there is no change in the expression of COX-2. Except some writing organization in the abstract, the paper is well organized and presented. Suggestions are listed as follows:
In the abstract, the background part could be shorter, like the section of “Inflammation is a severe topic ……………….play a role as pro-inflammatory mediators.” Instead, since scutellarein was found to be active on inhibiting SARS coronavirus helicase, and existed in some Chinese medicine that could be used to against other inflammatory relevant disease as Covid-19, so it would be nesseray to verify the pathway of anti-inflammatory of scutellarein. Therefore, a sentence for the hypothesis or aim could be added in the abstract, such as: To elucidate the if SCU can be used as an inflammation inhibitor and the possible inhibition pathway, the scutellarein (SCU) effect on LPS stimulated RAW264.7 cells was investigated.
And then, the last sentence of “Therefore, this study shows that SCU can be used as a compound to treat inflammation.”, could be deleted.
Response : Thank you for your advice. I made changes to the introduction based on your advice.

Reviewer 2 Report
Authors have looked for proteins/enzymes which come downstream of the inflammatory pathway. Inflammation primarily depends on tissue injury followed by activation of PLC to release Ca2+ ions for the activity of sPLA2. The product of sPLA2 is the substrate for COX/LOX enzymes which drives the inflammation. So, if the possible author can look for the activity of sPLA2 with respect to SCU, which will give a full detailed picture of the SCU being anti-inflammatory. Because SCU is a flavonoid, it might inhibit sPLA2, as quercetin. Overall I feel that rather anti-inflammatory, its anti-cancer pharmacological property is driving SCU to inhibit the transcription factors which is common in many progression of cancer (already reported).
Author Response
Authors have looked for proteins/enzymes which come downstream of the inflammatory pathway. Inflammation primarily depends on tissue injury followed by activation of PLC to release Ca2+ ions for the activity of sPLA2. The product of sPLA2 is the substrate for COX/LOX enzymes which drives the inflammation. So, if the possible author can look for the activity of sPLA2 with respect to SCU, which will give a full detailed picture of the SCU being anti-inflammatory. Because SCU is a flavonoid, it might inhibit sPLA2, as quercetin. Overall I feel that rather anti-inflammatory, its anti-cancer pharmacological property is driving SCU to inhibit the transcription factors which is common in many progression of cancer (already reported).
Response: We sincerely appreciate your comments. We will check the upstream signal proteins(sPLA2, PLC) in further study by referring to your comments.

Reviewer 3 Report
1. This study was clear and interesting. However, the stability of the testing SCU need to be confirmed. How is the absorption of SCU in the cells? Because the stability of such kind of flavonoids is poor.
2. iNOS and COX-2 were measured and showed suppression. Were NO and PGE2 also measured?
Author Response
- This study was clear and interesting. However, the stability of the testing SCU need to be confirmed. How is the absorption of SCU in the cells? Because the stability of such kind of flavonoids is poor.
Response: Thank you for your comments. In a paper written on apigenin, a type of flavonoid, this paper shows that apigenin is absorbed by cancer cells and is effectively hydrolyzed in cancer cells (1). Therefore, even when flavonoids absorb even cancer cells, they show the stability of being absorbed without being degraded. It was also demonstrated that scutellarin has an anti-inflammatory effect in vivo (2). It has also been shown to be effective in animals and thus absorbency.
- iNOS and COX-2 were measured and showed suppression. Were NO and PGE2 also measured?
Response: Thank you for your comments. As a result of confirming NO and PGE2, it was confirmed that both had no effect, and specifically, iNOS was effective.
- Srivastava, Janmejai K., and Sanjay Gupta. "Extraction, characterization, stability and biological activity of flavonoids isolated from chamomile flowers." Molecular and cellular pharmacology 1.3 (2009): 138.
- Peng, Ling, et al. "Scutellarin ameliorates pulmonary fibrosis through inhibiting NF-κB/NLRP3-mediated epithelial–mesenchymal transition and inflammation." Cell death & disease11 (2020): 1-16.

Round 2
Reviewer 3 Report
The revised manuscript was well done.
Author Response
Thank you for your response. I took your advice and added it to the discussion section at 250 lines with a reference.
